# Unveiling the Genomic Basis of Chemosensitivity in Sarcomas of the Extremities: An Integrated Approach for an Unmet Clinical Need

**DOI:** 10.3390/ijms24086926

**Published:** 2023-04-08

**Authors:** Silvia Vanni, Valentina Fausti, Eugenio Fonzi, Chiara Liverani, Giacomo Miserocchi, Chiara Spadazzi, Claudia Cocchi, Chiara Calabrese, Lorena Gurrieri, Nada Riva, Federica Recine, Roberto Casadei, Federica Pieri, Ania Naila Guerrieri, Massimo Serra, Toni Ibrahim, Laura Mercatali, Alessandro De Vita

**Affiliations:** 1Preclinic and Osteoncology Unit, Biosciences Laboratory, IRCCS Istituto Romagnolo per lo Studio dei Tumori (IRST) “Dino Amadori”, 47014 Meldola, Italy; 2Clinical and Experimental Oncology, Immunotherapy, Rare Cancers and Biological Resource Center, IRCCS Istituto Romagnolo per lo Studio dei Tumori (IRST) “Dino Amadori”, 47014 Meldola, Italy; 3Biostatistics and Clinical Trials Unit, IRCCS Istituto Romagnolo per lo Studio dei Tumori (IRST) “Dino Amadori”, 47014 Meldola, Italy; 4Medical Oncology Unit, Azienda Ospedaliera “San Giovanni Addolorata”, 00184 Roma, Italy; 5General and Oncologic Surgery, “Morgagni-Pierantoni” Hospital, 47121 Forlì, Italy; 6Pathology Unit, “Morgagni-Pierantoni” Hospital, 47121 Forlì, Italy; 7Osteoncologia, Sarcomi dell’osso e dei tessuti molli, e Terapie Innovative, IRCCS Istituto Ortopedico Rizzoli, 40136 Bologna, Italy

**Keywords:** myxofibrosarcoma, undifferentiated pleomorphic sarcoma, genomics, chemosensitivity, patient-derived cultures

## Abstract

Myxofibrosarcoma (MFS) and undifferentiated pleomorphic sarcoma (UPS) can be considered as a spectrum of the same disease entity, representing one of the most common adult soft tissue sarcoma (STS) of the extremities. While MFS is rarely metastasizing, it shows an extremely high rate of multiple frequent local recurrences (50–60% of cases). On the other hand, UPS is an aggressive sarcoma prone to distant recurrence, which is correlated to a poor prognosis. Differential diagnosis is challenging due to their heterogeneous morphology, with UPS remaining a diagnosis of exclusion for sarcomas with unknown differentiation lineage. Moreover, both lesions suffer from the unavailability of diagnostic and prognostic biomarkers. In this context, a genomic approach combined with pharmacological profiling could allow the identification of new predictive biomarkers that may be exploited for differential diagnosis, prognosis and targeted therapy, with the aim to improve the management of STS patients. RNA-Seq analysis identified the up-regulation of MMP13 and WNT7B in UPS and the up-regulation of AKR1C2, AKR1C3, BMP7, and SGCG in MFS, which were confirmed by in silico analyses. Moreover, we identified the down-regulation of immunoglobulin genes in patient-derived primary cultures that responded to anthracycline treatment compared to non-responder cultures. Globally, the obtained data corroborated the clinical observation of UPS as an histotype refractory to chemotherapy and the key role of the immune system in determining chemosensitivity of these lesions. Moreover, our results confirmed the validity of genomic approaches for the identification of predictive biomarkers in poorly characterized neoplasms as well as the robustness of our patient-derived primary culture models in recapitulating the chemosensitivity features of STS. Taken as a whole, this body of evidence may pave the way toward an improvement of the prognosis of these rare diseases through a treatment modulation driven by a biomarker-based patient stratification.

## 1. Introduction

Soft tissue sarcoma (STS) encompasses a wide range of solid malignancies exhibiting different biological and clinical behavior, which could vary from indolent to very aggressive [1,2]. In this regard, a clear example is represented by myxofibrosarcoma (MFS) and undifferentiated pleomorphic sarcoma (UPS), the most common STS of the extremities in adults and the fourth most common STS, respectively [3]. These mesenchymal-derived lesions were previously classified as malignant fibrous histiocytoma (MFH) until 2002, and then MFS was recognized as a separate entity by WHO in 2020 [4]. Nevertheless, MFS and UPS are often considered a spectrum of the same disease entity. Indeed, MFS exhibits a strong predilection for extremities, with low metastatic potential but extremely high rates of local recurrences, which may ultimately lead to limb amputation. On the other hand, UPS can arise in any part of the body, and it represents an aggressive mesenchymal disease prone to distant recurrence and metastasis, being therefore correlated with poor prognosis.

In this regard, MFS is characterized by curvilinear vessels, myxoid stroma, and pleomorphism, which currently represent the key hallmarks in the standard differential diagnosis. As a result, a major diagnostic pitfall is represented by the unavailability of specific immunohistochemical biomarkers and the presence of highly complex karyotypes [5]. UPS is characterized by a transition from storiform to pleomorphic areas representing a highly variable morphology. Since no specific diagnostic biomarkers are currently available, the diagnosis is mainly based on the exclusion of other cancer histotypes e.g., dedifferentiated and pleomorphic liposarcoma, pleomorphic leiomyosarcoma, pleomorphic rhabdomyosarcoma, MFS, poorly differentiated carcinoma, and melanoma [6]. The standard of care for localized STS including MFS and UPS is represented by surgery combined with (neo)adjuvant radio and chemotherapy in selected cases. In advanced and metastatic diseases, the gold standard is represented by chemotherapy with a very poor outcome. Anthracycline-based regimens represent the up-front chemotherapy treatment with a response rate ranging from 16% to 27% [7]. Second-line treatments may include a variety of chemotherapeutics (e.g., trabectedin, pazopanib, eribulin, gemcitabine-based regimens, high-dose ifosfamide, dacarbazine), among which the optimal regimen has not yet reached a general consensus [8].

Although several differences exist between the two histotypes, these solid lesions suffer, as for all STS, from the unavailability of effective biomarkers able to predict treatment response, either for the neo(adjuvant) or metastatic settings. Therefore, there is a pressing need to identify new predictive tools able to discriminate between responder and non-responder patients, in order to better define the best therapeutic options for each patient [9]. The recent advent of next generation sequencing (NGS) techniques represents a powerful tool to deepen the understanding of the natural history of these mesenchymal lesions (e.g., fusion transcripts as SS18-SSX, FUS-CHOP) together with the identification of biomarkers for targeted therapy (e.g., rearrangements of ALK for crizotinib and NTRK for larotrectinib and entrectinib) [10]. In this context, an integrated approach combining both genomic and pharmacological profiling would definitely provide a step forward in improving the management of STS patients, allowing clinicians to identify the right treatment for each patient (Figure 1). This pilot study pointed to select potential predictive biomarkers for MFS and UPS management in order to stratify patients undergoing surgical treatment, with the aim to better indicate those which could really benefit from chemotherapy treatment.

## 2. Results

### 2.1. Patients

Clinical–pathological characteristics of patients are reported in Table 1. Representative images of MFS and UPS patient surgical specimens as well as sagittal and axial MRI images are available in the Appendix A. MFS MRI images show a highly intense mass signal a typical of myxoid matrix in the subcutaneous tissues and a tail-like margin at caudal extent of lesion. UPS MRI images show an extensive palpable mass in the left thigh, well-defined by a pseudocapsule and characterized by an infiltrative pattern in the subcutaneous and muscular tissue. Representative images of H&E staining of the patients’ surgically resected tumor specimens are provided in Figure 1 (upper panels). Representative images of the IHC biomarkers detected in analyzed case series are provided in (Appendix A).

### 2.2. Establishment of Patient-Derived MFS and UPS Primary Cell Lines

Establishment of 12 MFS and 16 UPS patient-derived primary cultures was achieved. The cell lines continued to proliferate after ten culture passages. Morphological analyses were carried out on patient-derived isolated cells cultured within collagen-based scaffolds (Figure 1, lower panels). An experienced sarcoma pathologist confirmed the establishment of MFS and UPS primary cultures through assessment of hematoxylin and eosin-stained primary cultures, following the WHO [1]. We observed that the combination of 3D culture-based systems and patient-derived primary cultures was partially able to recapitulate some morphological features of growing MFS and UPS tumors. In this regard, atypical cells and bizarre mitotic features were detected.

### 2.3. Chemobiogram Assessment

Prospective study focusing on a case series of 28 MFS and UPS patient-derived primary cell cultures exposed to epirubicin-based regimen showed an intra-histotype stratification in responder and non-responder tumors. In particular, the selected arbitrary threshold of 50% cell viability identified 14 NR primary cultures, 6 MFS and 8 UPS. On the other hand, a total of 14 R primary cultures were observed, 6 MFS and 8 UPS (Figure 2A). The UPS cell lines chemosensitivity to epirubicin ranged from 11% to 92% of cell survival, while the MFS cell lines viability ranged from 22% to 84%. Moreover, the average cell survival of cells exposed to epirubicin was 55.3% for MFS against 48.2% for UPS. Furthermore, the bright-field microscopy images of representative MFS and UPS cell lines exposed to epirubicin showed a highly heterogeneous morphology, not only inter-histotype but also intra-histotype (Figure 2B). Indeed, atypical cell shapes and bizarre mitotic figures were detected in both untreated groups. Moreover, widespread debris together with apoptotic bodies were observed in the R cultures, while the NR cultures retained the overall cell architecture after drug exposure.

### 2.4. RNA Sequencing Profiling

First, we performed Principal Component Analysis (PCA) for quality assessment and exploratory analysis, with the aim to identify clusters of closely related data points (Appendix A).

When considering all 19 samples, those with lower NGS library quality and poor read yield clustered separated from the others, as shown in Appendix A.

These were therefore excluded from subsequent analyses, using a cut-off value of 4 M of total aligned reads (Appendix A). When considering the remaining 14 samples, no clear clustering of samples according to histotype or responsiveness condition (R vs. NR) was observed (Appendix A). However, samples were partially clustered according to gender (F vs. M) (Appendix A). We also performed hierarchical clustering of samples according to the top 500 most variable genes (Figure 3). Similar to what was observed in PCA, the heatmap showed that the F and M samples somewhat clustered separately as well as high- and low-reads samples. Moreover, replicates of the same samples prepared in different library batches and sequenced in different runs clustered together (samples 000001, 000002 and 000003), ensuring robustness of different library preparations across time. Taken together, both PCA and hierarchical clustering analyses identified the sex and number of aligned reads as possible confounding variables and were consequently included in the statistical models fitted for the differential gene expression tests. In particular, the correction for the number of aligned reads was performed with the aim of limiting the bias due to RNA degradation from FFPE.

Differential expression analysis (DEA) was performed comparing six MFS vs. eight UPS patients. The analysis identified 164 differentially expressed genes (DEGs) (log2FC ≤ |2| and *p* adj ≤ 0.05), of which 56 were up-regulated and 108 were down-regulated in UPS compared to MFS (Appendix A). In more detail, collagen assembly and formation genes (COL11A1, COL10A1) and cell migration gene MMP13 were significantly up-regulated in UPS cases, as well as human leukocyte antigen gene HLA-DQB1 and WNT7B gene. Conversely, cadherin genes (PCDH20, CDH19) cytochrome P450 genes (CYP2C18, CYP4Z1, CYP4F12, CYP26B1) and aldo/keto reductase superfamily genes (AKR1C2, AKR1C3) were down-regulated in UPS, as well as NTRK2, BMP7 and SGCG genes. Moreover, Gene Set Enrichment Analysis showed the down-regulation of several pathways in MFS compared to UPS such as membrane trafficking, vesicle-mediated transport, transcription. In particular, the most interesting pathways were neutrophil activation and cellular protein metabolic processes (Figure 4). We also found the up-regulation of doxorubicin metabolic processes in MFS compared to UPS.

Next, DEA was performed only considering patients whose derived primary cultures were treated with epirubicin. In detail, two responder (LR) vs. four non-responder (LNR) patients were analyzed. The analysis identified 61 DEGs (log2FC ≤ |2| and *p* adj ≤ 0.05) of which 41 were up-regulated and 20 were down-regulated in LR compared to LNR patients (Appendix A). In more detail, the immunoglobulin genes (IGKV2D-30, IGKV1D-13, IGHV3-72, IGLV3-10, IGHV1-69-2, IGKV3D-15) were down-regulated in LR patients compared to LNR ones. On the other hand, the desmin gene (DES) was up-regulated as well as the keratin genes (KRT10, KRT 80, KRT15). Moreover, GSEA showed the down-regulation of several pathways in LR compared to LNR, such as neutrophils mediated immunity and activation, class I MHC mediated antigen processing, mitotic cell cycle, and negative regulation of apoptotic process. Up-regulated pathways in LR were cell adhesion and calcium binding.

### 2.5. Analyses of Public Databases

We performed in silico analyses for MMP13, WNT7B, AKR1C2, AKR1C3, BMP7 AND SGCG genes on Sarcoma (TCGA, PanCancer Atlas) database including 50 UPS and 25 MFS patients and Adult Soft Tissue Sarcomas (TCGA, Cell 2017) database including 44 UPS and 17 MFS, and we found that in both studies the median mRNA expression values of MMP13 and WNT7B were higher in UPS patients compared to MFS ones, while the median values of AKR1C2, AKR1C, BMP7 and SGCG were higher in MFS patients compared to UPS ones (Figure 5 and Appendix A).

## 3. Discussion

Despite the improvement in the understanding of molecular biology and clinical behavior of STS, the limited availability of large clinical trials together with their poor outcome compared to some other solid malignancies pushes the need of deepening the pathophysiology of these mesenchymal entities. In this regard, an example is the case of MFS and UPS representing two separate STS entities previously grouped under the name of malignant fibrous histiocytoma that are both characterized by the unavailability of both diagnostic and prognostic biomarkers and generally harbor highly complex karyotypes [11,12]. Moreover, the lack of predictive biomarkers that could guide physicians in the therapy selection represents a critical issue in the clinical management of STS, including these two entities [13]. In this work, we took advantage of an integrated approach combining the translational strength of patient-derived primary cultures as a powerful tool for the study of pathophysiology and of the pharmacological profile of STS, together with genomic profiling techniques in order to identify promising biomarkers for the stratification of MFS and UPS patients for the most suitable chemotherapeutic treatment. In particular, in this proof-of-concept study, we investigated intra- and inter-genomic variability of these two histotypes, which could be in part responsible for different chemo-susceptibility and chemoresistance.

First, our results showed the ability of patient-derived primary cultures in resembling some of the morphological features of MFS and UPS. In this regard, the cytological analysis performed by experienced sarcoma pathologists on established primary cultures seeded within a 3D culture-based system confirmed the diagnosis of MFS and UPS tumors (Figure 1, lower panels). Next, to investigate the efficacy of gold standard anthracycline-based chemotherapeutic regimen, we exposed MFS and UPS patient-derived culture case series to epirubicin. Concerning the pharmacological profiling (Figure 2A), overall, the UPS chemosensitivity to epirubicin was extremely widespread, thus confirming the great heterogeneity of these entities, which is reflected in the challenging clinical management [14]. On the other hand, the average median chemosensitivity of MFS cell lines was lower compared to UPS ones (55% vs. 48%), confirming the partial chemo-refractoriness of these malignancies as observed in the daily clinical practice. These data also provide evidence about the ability of our patient-derived primary culture model to recapitulate the sensitivity of these entities to anthracyclines, as well as their histologic features. Indeed, heterogeneous morphology of these poorly differentiated lesions was reflected by the observation of atypical cell shapes and bizarre mitotic figures in both the MFS and UPS cell lines (Figure 2B). Moreover, the sensitivity of the R cultures to anthracyclines was confirmed by the presence of cell debris and apoptotic bodies that are likely related to the mechanism of action of epirubicin, which induces double-strand DNA breaks. Taken together, our results also highlight the loose correlation between histotype definition itself and the chemosensitivity, reinforcing the need to identify predictive biomarkers regardless of the histotype of the lesion.

Concerning RNA-Seq analyses, they revealed the up-regulation of WNT7B and BMP7 in UPS patients. WNT7B is a member of the WNT gene family whose altered signaling has been already related with oncogenesis in sarcomas [15]. Moreover, the positive regulation of the Wnt/β-catenin signaling pathway is known to maintain a primitive stem-like state and it has been recently shown that BMP signaling promotes EMT in osteosarcoma cells associated with enhanced motility and invasiveness through the Wnt/β-catenin signaling pathway [16]. Therefore, these data are suggestive of poor differentiation and increased cell migration, which are consistent with the clinical–pathological features of UPS. Moreover, even though the success of the blockade of Wnt signaling as a cancer treatment has been hampered by consequent severe side effects such as alteration of tissue homeostasis and regeneration, recently several Wnt signaling inhibitors have been developed and are currently being used in clinical trials, opening new therapeutic options also for STS [17].

In addition, we also reported the up-regulation of the genes involved in collagen and matrix remodeling such as COL11A1, COL10A1, and MMP13, which are associated with altered ECM composition, likely favoring cell invasiveness and aggressiveness [18]. Moreover, we observed the up-regulation of HLA gene HLA-DQB1, as well as neutrophils activation and MHC antigen processing in UPS cases compared to MFS, highlighting the involvement of the immune system and providing the rationale for ICI-based therapeutic strategies in this histotype. On the other hand, we detected the down-regulation of cell adhesion (PCDH20, CDH1) and the Cytochrome P450 genes as well as unbalanced mitochondrial activity in UPS cases compared to MFS, which has been previously correlated with poor prognosis and induction of EMT and therefore metastatic propensity [19]. Interestingly, we also identified the down-regulation of NTRK2 in UPS cases. Rearrangements of the NTRK genes have been described in several soft tissue cancer types, and their relevance is mainly due to the availability of targeted TRK inhibitors, which showed great efficacy in triggering tumor shrinkage [20]. Besides this, NTRK2 is able to phosphorylate itself and several members of the MAPK pathway signaling, leading to cell differentiation. Moreover, we detected the down-regulation of the SGCG gene that encodes gamma-sarcoglycan, one of several sarcolemmal transmembrane glycoproteins that interact with dystrophin. The dystrophin-glycoprotein complex (DGC), which comprises sarcoglycans, provides a structural link between the subsarcolemmal cytoskeleton and the extracellular matrix of muscle cells. Therefore, their down-regulation in UPS could help explain the poor differentiation typical of these lesions. Last, we observed the up-regulation of the AKR1C2 and AKR1C3 genes and of the doxorubicin metabolism processes in MFS patients compared to UPS. Aldo-keto reductase family members deregulation, and in particular AKR1C3, has been correlated to poor prognosis likely due to inactivation of anthracyclines metabolism [21,22]. This is in line with our findings about increased average cell survival of MFS-derived primary cultures treated with anthracycline compared to UPS-derived cultures. These data further support the reliability of our patient-derived culture system as a robust in vitro chemoresponse model for these STS histotypes. Moreover, we confirmed our RNA-Seq results in a wider case series of 50 UPS and 25 MFS patients using the Sarcoma (TCGA, PanCancer Atlas) database and in a case series of 44 UPS and 17 MFS patients using the Adult Soft Tissue Sarcomas (TCGA, Cell 2017) database for MMP13, WNT7B, AKR1C2, AKR1C3, BMP7, AND SGCG genes. Indeed, in silico analyses conducted on both datasets corroborated our findings, further highlighting the robustness of our pipeline.

Next, given the interesting results that emerged about anthracycline chemoresistance in about half of the treated cell lines in both histotypes, and considering the limited availability of CT data from these patients who typically only undergo surgery as a first-line treatment, we decided to perform an additional analysis comparing patients whose primary cell lines were responsive to epirubicin (LR) versus patients whose cell lines were non-responders (LNR). Interestingly, we identified the down-regulation of several immunoglobulin genes as well as down-regulation of neutrophil-mediated immunity and activation pathways in LR patients compared to LNR ones. These results are particularly interesting in light of recently published results, which showed that an increased neutrophils-to-lymphocyte ratio was significantly associated with worse PFS in a case series of 99 STS patients [6]. Indeed, it is known from the literature that neutrophils can remodel the extracellular matrix and promote angiogenesis, thereby stimulating tumor cell migration and metastasis. Furthermore, neutrophils have a significant impact on immunity by inhibiting the cytolytic activity of lymphocytes, while tumor-infiltrating lymphocytes can limit the metastatic growth of tumor cells [23,24].

Some limitations are present in this study, mainly represented by FFPE samples that are characterized by highly fragmented nucleic acids that do not allow the isolation of high quality RNA, significantly affecting library preparation and sequencing depth. However, dealing with rare tumors, it is common practice to use archival samples embedded over decades to allow sufficient sample size for reliable studies. However, in order to minimize the effect of low quality RNA, we excluded from the analyses samples showing very low read number and applied correction variables to the included samples as well as stringent FC criteria, thereby increasing the robustness of our results. Moreover, the small case series and the limited responder and non-responder sample size represent one of the major drawbacks in the study of rare tumors, which we attempted to partially overcome by confirming our results with wider cases series from publicly available databases. In this regard, we are planning to further improve our case series and to validate the obtained results via bioinformatic platforms such as Connectivity Map (CMap), which can be exploited to discover the mechanism of action of small molecules, functionally annotate genetic variants of disease genes, and inform clinical trials [25,26,27].

## 4. Materials and Methods

### 4.1. Case Series

The study enrolled 39 sarcoma patients (UPS = 20 and MFS = 19) surgically resected by experienced orthopedic surgeons. The explanted tumor masses were analyzed by a sarcoma pathologist and processed within 3 h of surgical resection. Clinical–pathological characteristics of patients (sex, age at treatment, histotype, adjuvant chemotherapy, adjuvant radiotherapy, tumor burden) are reported in Table 1. IRST-Area Vasta Romagna Ethics Committee approved the study protocol, approval no. 4751, 31 July 2015. Good Clinical Practice standard operating procedures and 1975 Helsinki declaration were applied in the study. All human samples were anonymized. Enrollment of patients started in September 2015 and ended in July 2022.

### 4.2. Collagen-Based Scaffold 3D Culture Model Synthesis

Synthesis of tridimensional collagen-based scaffold culture systems was performed in our laboratory as previously reported [28]. Briefly, bovine-derived type I microfibrillar collagen (Merck KGaA, Darmstadt, Germany) was dispersed in acetic acid solution (Merck KGaA, Darmstadt, Germany) with the addition of sodium hydroxide solution (Merck KGaA, Darmstadt, Germany) and 1,4-butanediol diglycidyl ether (BDDGE) solution (Merck KGaA, Darmstadt, Germany) a crosslinker. The resulting suspension was then homogenized (T18 Basic ULTRA-TURRAX, IKA^®^ Werke GmbH & Co. KG, Staufen, Germany) and freeze-dried with a controlled freezing and heating ramp under vacuum conditions (Labconco Corporation, Kansas City, MO). Scaffolds were then sterilized in ethanol 70% for 1 h and washed with PBS before use.

### 4.3. Histologic Characterization of Tumor Specimens and Derived Primary Cell Lines

For analysis of histopathological features, tumor tissues and their corresponding patient-derived cell lines grown in 3D collagen-based scaffolds were fixed in 10% neutral buffered formalin and then embedded in paraffin. As previously described by our group [29], 5 µm sections were cut and flattened on a heated water bath, floated onto microscope slides, and dried. Slides were then stained with hematoxylin and eosin (H&E). Microscopic analysis of H&E staining was conducted in a blind fashion by a board-certified pathologist.

### 4.4. Isolation of Patient-Derived UPS and MFS Cell Lines

MFS and UPS patient-derived primary cells were isolated and established according to the protocol previously reported [29]. Briefly, surgical-resected tumor tissue was washed in PBS and finely crumbled with surgical scalpels. The resulting tumor fragments were enzymatically digested with collagenase type I (Millipore Corporation, Billerica, MA, USA) supplemented with 1:1 Dulbecco’s-modified Eagle’s medium (DMEM, Invitrogen, Darmstadt, Germany) for 15 min at 37 °C under stirring conditions. The sample was then stored overnight under stirring conditions at room temperature. The day after, enzymatic digestion was stopped by adding DMEM supplemented with 10% fetal bovine serum (Invitrogen, Darmstadt, Germany), 1% penicillin/streptomycin, and 1% glutamine. The cell suspension was then filtered, and the isolated cells were counted and seeded in standard monolayer cultures with a cell density of 80,000 per cm2. Primary cultures were maintained in DMEM supplemented with 10% fetal bovine serum (Invitrogen, Darmstadt, Germany), 1% penicillin/streptomycin, and 1% glutamine at 37 °C in a 5% CO2 atmosphere. For establishing effective tumor cell isolation, after ten passages cell cultures were seeded onto collagen-based scaffolds with a cell density of 500,000 cells/mm3 and grown for 72 h. All the experiments were conducted using low-passage and proliferating primary cultures.

### 4.5. Pharmacological Profiling

The efficacy of in vitro epirubicin treatment was assessed by MTT (3-(4,5-dimethylthiazol-2-yl)-2,5-diphenyltetrazolium bromide) reduction assay. MTT salt was purchased from Sigma-Aldrich (St. Louis, MO, USA) and dissolved in PBS (5 mg/mL). Briefly, tumor cells were seeded in 96-well plates at a density of 80,000 cells/cm2. Cells were allowed to attach for 24 h before drug exposure and cell survival percentage was assessed after 72 h of drug exposure. Culturing media was replaced with the MTT solution for 2 h at 37 °C. Then, MTT solution was replaced with 0.01 M HCl in isopropanol for crystals solubilization. Next, 96-well plates were measured using a Synergy H1 BioTek microplate reader (Agilent Technologies, Santa Clara, CA, USA) at 550 nm wavelength. The regimen was selected according to peak plasma concentration 3.4 µg/mL (Accord Healthcare Italia Ltd., Milan, Italy) of epirubicin from pharmacokinetic clinical data [30] and used in previously published translational research studies [31]. The experiments were performed twice with eight replicates for each condition. Patient-derived primary cultures stratification in responder (R) and non-responder (NR) groups was performed using an arbitrary cut-off of 50% of cell survival after 72 h of drug exposure (R > 50, NR < 50).

### 4.6. Tissues and RNA Extraction

Total RNA was isolated from 19 (MFS n = 10 and UPS n = 9) surgically resected FFPE tissue sections, after checking of the tissue representativeness, using the RNeasy FFPE kit (Qiagen). Total RNA concentration was measured using Nanodrop (Thermo Fisher Scientific, Waltham, MA, USA) and the quality was checked on the 2100 Bioanalyzer with the RNA 6000 Pico kit (Agilent Technologies, Santa Clara, CA, USA).

### 4.7. Next Generation Sequencing Analysis

Paired-end RNA sequencing was performed as follows: library preparation was performed with Hamilton NGS STAR using Illumina Stranded Total RNA Prep, Ligation with Ribo-Zero Plus (Illumina) according to manufacturer’s instructions. Library concentration was measured using the Qubit dsDNA BR assay kit (Thermo Fisher Scientific, Waltham, MA, USA) and the quality was assessed on the 2100 Bioanalyzer with the DNA 1000 kit (Agilent Technologies, Santa Clara, CA, USA). Libraries were then denatured and diluted to a final concentration of 1.6 pM and subsequently run on the Illumina NextSeq 550 (Illumina) platform using NextSeq 500/550 High Output Kit v2.5 (150 Cycles) according to the manufacturer’s instructions.

### 4.8. Bioinformatic NGS Data Analysis

Transcript-level read count was performed with Kallisto v0.46.2. Raw counts were collapsed to gene-level with tximport v1.12.1, and then Differential Expression Analysis (DEA) was performed with DESeq, version v1.22.1 (Bioconductor, version 3.16, https://bioconductor.org) [32,33,34]. The same software DESeq, version v1.22.1 (Bioconductor, version 3.16, https://bioconductor.org) was used to produce PCA plots, while for heatmaps and hierarchical clustering seaborn v0.12.1 was used. Gene Set Enrichment Analysis (GSEA) was performed with the package GSEApy v0.9.16 (https://pypi.org/project/gseapy/) [35,36,37]. For QC analyses, DESeq2’s Variance Stabilizing Transformation (VST) was applied to raw read counts, and the 500 genes with highest variance across the dataset were selected to perform PCA and hierarchical clustering; VST-transformed counts were also used as input for GSEA, after removing genes <1 across all samples. GSEA were analyzed using three different genomic repositories, i.e., Kyoto Encyclopedia of Genes and Genomes (KEGG) Human Pathway Database v. 2019, Reactome v. 2016 and Gene Ontology (GO) Biological Process, v. 2018. All analyses were performed comparing MFS vs. UPS or patients with responder cell lines (LR) vs. patients with non-responder cell lines (LNR). RNA-seq data that support the findings of this study are available upon reasonable request.

### 4.9. In Silico Analysis on Public Clinical Dataset

Datasets were derived from the web-accessible database cBio Cancer Genomics Portal (https://www.cbioportal.org/ accessed on 15 February 2023) [38,39]. Sarcoma (TCGA, PanCancer Atlas) study including 253 patients with mRNA expression data and Adult Soft Tissue Sarcomas (TCGA, Cell 2017) study including 206 patients with mRNA expression data were analyzed.

### 4.10. Statistical Analysis

Concerning pharmacological profiling, three independent replicates were performed for each experiment. Data are presented as mean ± standard deviation (SD) or standard error (SE), as stated. Differences between groups were assessed by a two-tailed Student’s t-test and accepted as significant at *p* < 0.05. Concerning NGS analyses, Python 3.6.5 and R 3.5.1 were used for statistical analyses.

## 5. Conclusions

The molecular determinants of chemosensitivity in STS are not well recognized and there is no tight correlation with histotype. Upon further validation in a wider cohort, our results could help in providing predictive biomarkers that can be used in clinical practice as tools for stratification of patients in order to identify the best therapeutic option, with the ultimate goal of improving the management of these challenging diseases.

## Data Availability

The datasets generated and/or analyzed during the current study are available from the corresponding author on reasonable request.

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
