# Peer review of "Unveiling the Genomic Basis of Chemosensitivity in Sarcomas of the Extremities: An Integrated Approach for an Unmet Clinical Need"

_ijms, 2023, doi:10.3390/ijms24086926_

Round 1
Reviewer 1 Report
The manuscript made by Vanni S et al., is interesting, well done, and well explained.
The manuscript describes in detail the features of UPS and MFS, comparing responder and non-responder tumors. The authors explain several characteristics of both tumors (clinical data, images, histological, immunohistochemistry, molecular data, cell culture). The introduction has an adequate extension, is updated, and is well explained. The authors focused on giving data about their study.
The material and methods were extensive and explained in detail.
However, they need to give information about the ethics committee authorization, use standard deviation in the age variable, explain in Table 1 the other types of treatment, not only chemotherapy, and explain how many females and males had UPS and MFS (F.E.: Sarcoma by sex, Female UPS: 10, Female MFS: 9, Male, etc.).
Explain if the pathologist used the WHO classification of soft tissue and bone tumors.
Overall comments
This is a well-written manuscript that will be an important manuscript in the future. The authors provide important data about two of the most important STS.
Author Response
We thank Reviewer 1 for the useful comments and suggestions which greatly improved our manuscript.
Below a point-by-point reply to the comments.
REVIEWER 1
The manuscript made by Vanni S et al., is interesting, well done, and well explained.
The manuscript describes in detail the features of UPS and MFS, comparing responder and non-responder tumors. The authors explain several characteristics of both tumors (clinical data, images, histological, immunohistochemistry, molecular data, cell culture). The introduction has an adequate extension, is updated, and is well explained. The authors focused on giving data about their study.
The material and methods were extensive and explained in detail.
However, they need to give information about the ethics committee authorization, use standard deviation in the age variable, explain in Table 1 the other types of treatment, not only chemotherapy, and explain how many females and males had UPS and MFS (F.E.: Sarcoma by sex, Female UPS: 10, Female MFS: 9, Male, etc.).
We thank Reviewer 1 for the suggestions provided. We added relevant information about Ethical Committee authorization in Methods section, paragraph 4.1 (highlighted in yellow) as follows: “IRST-Area Vasta Romagna Ethics Committee approved the study protocol, approval no. 4751, 31 July 2015. Good Clinical Practice standard operating procedures and 1975 Helsinki declaration were applied in the study.”
Concerning standard deviation of age at treatment, this was 69.19±13.6. We modified it accordingly in the manuscript (Table 1, highlighted in yellow).
Concerning other types of treatment besides chemotherapy, all the included patients received surgery as first-line treatment, followed by chemotherapy in 4 cases, as already mentioned in Table 1. 9 patients received adjuvant radiotherapy. None of the patients received pre-operative treatments. We modified Table 1 accordingly, including the number of males and females cases with UPS or MFS (highlighted in yellow).
Explain if the pathologist used the WHO classification of soft tissue and bone tumors.
We thank Reviewer 1 for the comment. The pathologist did follow WHO guidelines 5th Edition 2020 for classification of these lesions. We added this detail in the Methods section, paragraph 2.2 (highlighted in yellow).
Overall comments
This is a well-written manuscript that will be an important manuscript in the future. The authors provide important data about two of the most important STS.
We thank Reviewer 1 for these favourable comments.
Reviewer 2 Report
The authors performed an integration approach combining both genomic and pharmacological profiling, which would definitely provide a step forward in improving the management of STS patients, allowing clinicians to identify the right treatment for each patient. Meanwhile, the pilot study pointed to selecting potential predictive biomarkers for MFS and UPS management in order to stratify patients undergoing surgical treatment, with the aim to better indicate those who could really benefit from chemotherapy treatment. Albeit, I consider these findings to provide new insight into cancer-related fields, I still have some suggestions.
1, Most figures and tables are highly professional; however, the authors should guide the readers to the meaning of the images and tables appropriately; otherwise, it is likely to cause misunderstandings. Therefore, I suggest the author consider revising these figures and table legends again.
2, In Figure 3, the author used heatmap displaying the hierarchical clustering and heatmap of the 500 most variable genes. However, it would be much better if the author could label important genes in this plot. Meanwhile, the plot seems to have some problems with data processing, please check the data or code carefully for errors. I suggest they can try some bioinformatics tools for data visualization: http://www.bioinformatics.com.cn/srplot
3, Since Connectivity Map (CMap) can be used to discover the mechanism of action of small molecules, functionally annotate genetic variants of disease genes, and inform clinical trials. It would be fascinating if these data could be correlated with other clinical databases. Therefore, I suggest the authors can validate their data via CMap or proteinatlas, and discuss these methodologies and literature as well as the validated data for cancer recurrence or metastasis in the manuscript (PMID: 25613900, 29195078, 32064155)
4, The author paves the way toward an improvement of the prognosis of these rare diseases through a treatment modulation driven by a biomarker-based patient stratification. However, It would be much better if the authors could provide some Workflow or Scheme for this research, I suggest that they can take a look at the recent paper in MDPI (PMID: 24619302, 34834441)
5, The resolution of supplementary Figure 1 was quite poor, please improve these data for revision. Meanwhile, there are few typo issues for the authors to pay attention to; please also unify the writing of scientific terms. “Italic, capital”?
Author Response
We thank Reviewer 2 for the useful comments and suggestions which greatly improved our manuscript.
Below a point-by-point reply to the comments.
REVIEWER 2
The authors performed an integration approach combining both genomic and pharmacological profiling, which would definitely provide a step forward in improving the management of STS patients, allowing clinicians to identify the right treatment for each patient. Meanwhile, the pilot study pointed to selecting potential predictive biomarkers for MFS and UPS management in order to stratify patients undergoing surgical treatment, with the aim to better indicate those who could really benefit from chemotherapy treatment. Albeit, I consider these findings to provide new insight into cancer-related fields, I still have some suggestions.
1, Most figures and tables are highly professional; however, the authors should guide the readers to the meaning of the images and tables appropriately; otherwise, it is likely to cause misunderstandings. Therefore, I suggest the author consider revising these figures and table legends again.
We thank Reviewer 2 for the suggestions provided. We did our best to extensively describe figures and tables in each legend and in the manuscript. We carefully reviewed all the figures and table captions and improved their descriptions in order to help the readers in grasping the meaning (highlighted in yellow).
2, In Figure 3, the author used heatmap displaying the hierarchical clustering and heatmap of the 500 most variable genes. However, it would be much better if the author could label important genes in this plot. Meanwhile, the plot seems to have some problems with data processing, please check the data or code carefully for errors. I suggest they can try some bioinformatics tools for data visualization: http://www.bioinformatics.com.cn/srplot
We thank Reviewer 2 for the comments. The 500 genes used for hierarchical clustering were selected because they had the highest variance across the whole sample cohort, regardless of the grouping chosen for differential expression analysis. These are the genes that carry the most information in terms of gene expression, thus they are the most suitable to represent the RNA expression of each sample as a whole. Therefore, it is not possible at this step to deem some of these genes as more important than others. As for the code, we have carefully double-checked again the whole code and the data, but we were not able to find anything apparently wrong. As mentioned in the manuscript, for data visualization we used seaborn v0.12.1 for generating heatmaps and hierarchical clustering (Waskom, M.L. Seaborn: statistical data visualization. JOSS, 2021, 6, 3021. https://doi.org/10.21105/joss.03021.)
3, Since Connectivity Map (CMap) can be used to discover the mechanism of action of small molecules, functionally annotate genetic variants of disease genes, and inform clinical trials. It would be fascinating if these data could be correlated with other clinical databases. Therefore, I suggest the authors can validate their data via CMap or proteinatlas, and discuss these methodologies and literature as well as the validated data for cancer recurrence or metastasis in the manuscript (PMID: 25613900, 29195078, 32064155)
We thank Reviewer 2 for the suggestion. Since this is a proof-of-concept study with a limited number of patients included, we are currently working on expanding the case series enrolling new patients and including additional Centers. Of course we will perform the suggested validation via CMap in a future study taking into consideration also extended follow-up data. However, we have added a specific paragraph in the Discussion section to discuss this future perspective analysis, including the suggested references (highlighted in yellow).
4, The author paves the way toward an improvement of the prognosis of these rare diseases through a treatment modulation driven by a biomarker-based patient stratification. However, It would be much better if the authors could provide some Workflow or Scheme for this research, I suggest that they can take a look at the recent paper in MDPI (PMID: 24619302, 34834441)
According to the Reviewer suggestion we have included a schematic diagram depicting the workflow we followed and the proposed clinical management for these patients (Scheme 1). We also added the following description: “Scheme 1: Graphic representation of the experimental workflow and of the proposed clinical management. Experimental workflow (left side): MFS and UPS patients typically undergo surgery as first treatment. Taking advantage of the resulting tumor specimen, we combine our translational model of patient-derived primary culture with pharmacology and genomic profiling in order to identify responder and non responder patients. Proposed clinical management workflow: taking advantage of the identified biomarkers, we could perform a patient stratification in order to select the best treatment for each patient.”
5, The resolution of supplementary Figure 1 was quite poor, please improve these data for revision. Meanwhile, there are few typo issues for the authors to pay attention to; please also unify the writing of scientific terms. “Italic, capital”?
We improved the quality of the supplementary figures and we carefully revised the manuscript to correct the grammar mistakes and unify the writing.